# Backward compatibility of whole genome sequencing data with MLVA typing using a new *MLVAtype* shiny application for *Vibrio cholerae*

Jérôme Ambroise[1]*, Léonid M. Irenge[1], Jean-François Durant[1], Bertrand Bearzatto[1], Godfrey Bwire[2], O. Colin Stine[3], Jean-Luc Gala[1]

**1** Center for Applied Molecular Technologies, Institute of Clinical and Experimental Research, Université catholique de Louvain, Brussels, Belgium, **2** Ministry of Health Uganda, Department of Community Health, Kampala, Uganda, **3** University of Maryland School of Medicine, Department of Epidemiology and Public Health, Baltimore, Maryland, United States of America

\* jerome.ambroise@uclouvain.be

## Abstract

### Background

Multiple-Locus Variable Number of Tandem Repeats (VNTR) Analysis (MLVA) is widely used by laboratory-based surveillance networks for subtyping pathogens causing foodborne and water-borne disease outbreaks. However, Whole Genome Sequencing (WGS) has recently emerged as the new more powerful reference for pathogen subtyping, making a data conversion method necessary which enables the users to compare the MLVA identified by either method. The *MLVAType* shiny application was designed to extract MLVA profiles of *Vibrio cholerae* isolates from WGS data while ensuring backward compatibility with traditional MLVA typing methods.

### Methods

To test and validate the *MLVAType* algorithm, WGS-derived MLVA profiles of nineteen *Vibrio cholerae* isolates from Democratic Republic of the Congo (n = 9) and Uganda (n = 10) were compared to MLVA profiles generated by an *in silico* PCR approach and Sanger sequencing, the latter being used as the reference method.

### Results

Results obtained by Sanger sequencing and *MLVAType* were totally concordant. However, the latter were affected by censored estimations whose percentage was inversely proportional to the k-mer parameter used during genome assembly. With a k-mer of 127, less than 15% estimation of *V. cholerae* VNTR was censored. Preventing censored estimation was only achievable when using a longer k-mer size (*i.e.* 175), which is not proposed in the SPAdes v.3.13.0 software.

**Data Availability Statement:** All NGS data are available from the European Nucleotide Archive (ENA, http://www.ebi.ac.uk/ena), available under

study accession number ERP114722, and from the Sequence Read Archive (SRA, https://www.ncbi.nlm.nih.gov/sra) under study accession number PRJNA439310.

**Funding:** The author(s) received no specific funding for this work.

**Competing interests:** The authors have declared that no competing interests exist.

## Conclusion

As NGS read lengths and qualities tend to increase with time, one may expect the increase of k-mer size in a near future. Using *MLVAType* application with a longer k-mer size will then efficiently retrieve MLVA profiles from WGS data while avoiding censored estimation.

## Introduction

Rapid molecular typing of pathogens associated with human and animal diseases has proven instrumental in the surveillance and control of infectious diseases [1, 2]. Pulsed field gel-electrophoresis (PFGE), which was long considered as the gold standard for molecular typing of pathogens associated with outbreaks, has been superseded by Multi-Locus Sequence Typing (MLST) or Multi-Locus Variable Number of Tandem Repeats (VNTR) Analysis (MLVA), and more recently by Whole Genome Sequencing (WGS) [3].

However, unlike MLVA, WGS analysis requires a specific expertise in bioinformatics and is not yet affordable in all developing countries where highly pathogenic diseases would make it the most useful and such a method would be the most needed. One can therefore be sure that MLVA and WGS subtyping will coexist in years to come, making necessary a methodology enabling end-users (*i.e.* researchers, clinicians, microbiologist and epidemiologists) to compare respective results.

Accordingly, *in silico* methods which extract low-throughput typing results (e.g., MLST or MLVA) from WGS data should be developed to enable users to compare subtyping results irrespective of the methodology and time of data acquisition. Both parameters are important when WGS data need to be compared with data generated before the WGS era.

Whereas the number of tandem repeats at different VNTR loci may theoretically be retrieved from WGS data, like currently done when extracting MLST from WGS, this was practically not considered feasible with MLVA because of a lack of accuracy of genomes assembly derived from Next Generation Sequencing (NGS) short reads [4]. Limited backward compatibility of WGS with MLVA is indeed notoriously due to failure to correctly assemble repetitive regions assessed by MLVA [5]. However, it is worth noting that an *in silico* PCR approach to type MLVA from WGS data was recently developed and evaluated for *Brucella* [6] (https://github.com/dpchris/MLVA) and *Salmonella* species (https://github.com/Papos92/MISTReSS).

When analyzing NGS data, the first step generally consists in assembling reads into longer contiguous sequences (contigs), which can then be interrogated using BLAST or other search algorithms. The production of high quality assemblies using bacterial genome assembler such as SPAdes [7] requires quality filtering and optimization of different parameters including k-mer size.

In the current paper, we describe a new tool (named *MLVAtype*) which enables users to extract MLVA profiles of *Vibrio cholerae* isolates from WGS data. We tuned the k-mer parameter used during genome assembly in order to assess its impact on the performance and limitations of *MLVAtype*. As a proof of concept, this new tool was applied on draft genomes of isolates associated with cholera outbreaks in two bordering countries, *i.e.* the Democratic Republic of the Congo (DRC) and Uganda. Results were compared to MLVA profiles generated by an *in silico* PCR approach and Sanger sequencing, the latter being used as the reference method.

## Materials and methods

### Sample description

Nine *V. cholerae* isolates were selected from a collection of isolates characterized in a recent study conducted between 2014 and 2017 in the DRC [8]. In addition, ten *V. cholerae* isolates collected between 2014 and 2016 in Uganda by G. Bwire and colleagues were selected based on their published data [9].

### Sanger-derived versus GeneScan-derived MLVA typing

Sanger-derived MLVA typing was performed by sequencing amplicons on both strands on the ABI 3130 GA, using the BigDye Terminator v1.1 cycle sequencing kit (Applied Biosystems, USA). Motif repeats were counted manually and translated into MLVA profiles. For Ugandan isolates, the fluorescently labeled amplified products were separated using a 3730xl Automatic Sequencer with the size determined from internal lane standards (LIZ600) by the GeneScan program (Applied Biosystems, Foster City, CA).

When MLVA profiles were generated according to the method proposed by Kendall et al. [10], the formula had to be modified to better fit the sequence length of the motif and the position of the primers (Table 1). It is of note that, the original calculation formula was used for the VCA0283 locus but with a modified reverse primer (AGCCTCCTCAGAAGTTGAG instead of the original reverse primer [GTACATTCACAATTTGCTCACC]). It is worth nothing that using the original reverse primer would require to adapt the formula.

### WGS-derived MLVA typing

**WGS and short reads assembly.** Whole genome assemblies of selected *V. cholerae* isolates, 9 from DRC and 10 from Uganda, were generated from paired-end 300 and 150 nt long reads, respectively. In brief, genomic DNA from DRC isolates was simultaneously fragmented and tagged with sequencing adapters in a single step using Nextera transposome (Nextera XT DNA Library Preparation Kit, Illumina, San Diego, CA, USA). Tagged DNA was then amplified with a 12-cycle polymerase chain reaction (PCR), cleaned up with AMPure beads, and subsequently loaded on a MiSeq paired-end 2 x 300 nt (MiSeq reagent kit V3 (600 cycles) sequence run. Raw genomic data were submitted to the European Nucleotide Archive (ENA, http://www.ebi.ac.uk/ena), and are available under study accession number ERP114722. For Ugandan *V. cholerae* isolates, data which were retrieved from the previous publication [7], were obtained as follows: libraries for Illumina sequencing were prepared from DNA fragmented with Covaris E210 (Covaris, Wolburn, MA) using the KAPA High Throughput Library Preparation Kit (Millipore-Sigma, St. Louis MO). The libraries were enriched and barcoded in ten cycles of PCR amplification with primers containing an index sequence. Subsequently, the libraries were sequenced using a 150 nt paired-end run on an Illumina HiSeq2500

**Table 1. Formula used in the current study to compute the number of tandem repeats from the amplicon size.** *: Modifications introduced in the new formula.

| Loci | Motif | Formula used by Kendall et al. | New formula |
|---|---|---|---|
| VC0147 | aacaga | (X-150)/6 | (X-150)/6 |
| VC0437 | gacccta | (X-245)/6 | (X-252)/7* |
| VC1650 | ataatccag | (X-307)/9 | (X-308)/9* |
| VCA0171 | gctgtt | (X-270)/6 | (X-268)/6* |
| VCA0283 | ccagaa | (X-95)/6 | (X-95)/6 |

(Illumina, San Diego, CA). Raw genomic data were submitted to Sequence Read Archive (SRA, https://www.ncbi.nlm.nih.gov/sra) under study accession number PRJNA439310.

WGS data were assembled into contigs using SPAdes v.3.13.0 [7] and a range of k-mer sizes of 55, 77, 99, 127, and 175-mers. Testing a k-mer value of 175, which has a longer size than what is proposed in the SPAdes v.3.13.0 software, required editing the software source code. It has to be kept in mind that using this longer k-mer size is only possible with reads of sufficient length. Accordingly, a k-mer size of 175 was only used with 300 nt reads from DRC isolates, but not with 150 nt reads from Ugandan isolates. Alternatively, WGS data were also assembled using SPAdes but without specifying the k-mer size. For each sample, k-mer size and locus, the number of tandem repeats was extracted from the assembled draft genome using the *MLVA-type* algorithm.

***MLVAtype* algorithm.** The *MLVAtype* algorithm processes each locus separately. It requires several inputs including a draft genome, the size of the k-mer (*i.e.* k-mer parameter) used during genome assembly, and the nucleotide sequence of the motif. The algorithm returns the number of tandem repeats using the following steps: first, small contigs (< 1000 nt) are removed from the draft genome. Second, the vcountPattern function from the Bio-string R package is used to count the number of occurrences that a single (j = 1) motif is detected within the draft genome. Then the same computation is iteratively performed with an increasing number (j = 2, 3 .., k) of tandem repeats. This iterative process is performed until there is only one occurrence of the k tandem repeats. Finally, the maximum value of j (*i.e.* k) is compared to the maximum number of tandem repeats (MNTR) that can be included in a specified k-mer (Fig 1 and Table 2). If k ≥ MNTR, the estimation of the number of tandem repeats is set to MNTR and considered as right-censored (*i.e.* ≥MNTR). If k < MNTR, the estimation of the number of tandem repeats is set to k. The same process is applied on each locus in order to extract the complete MLVA profile from the genome assembly.

***MLVAtype* shiny application.** The *MLVAtype* algorithm was implemented in an R shiny application which is freely available at https://ucl-irec-ctma.shinyapps.io/NGS-MLVA-TYPING/. This application enables the user to upload a list of draft genomes, the nucleotide sequences of the motifs, and the value of the k-mer which was used to build the assembly, including a k-mer size selectable after modification of the SPAdes v.3.13.0 source code (Fig 2).

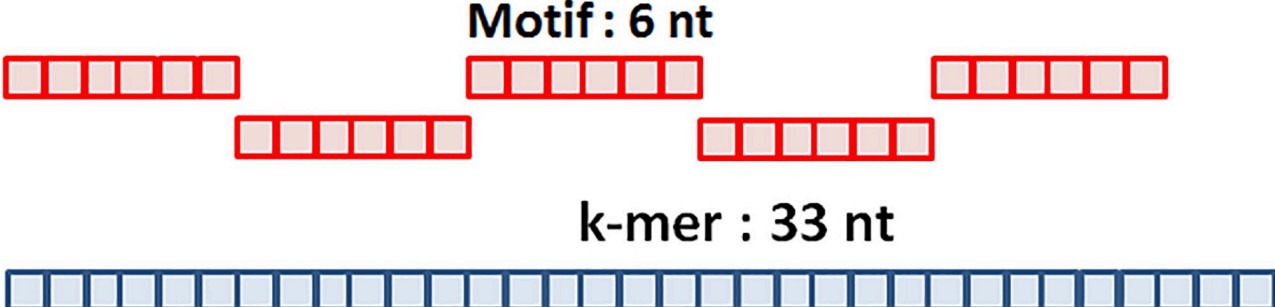

**Fig 1. Maximum number of tandem repeats (MNTR).** Example of the maximum number of a 6 nt motif included in a k-mer of 33 nt.

**Table 2. Maximum number of tandem repeats in a k-mer.**

| Loci | motif length | k-mer | | | | | |
|---|---|---|---|---|---|---|---|
| | | 33 | 55 | 77 | 99 | 127 | 175 |
| VC0147 | 6 | 5 | 9 | 12 | 16 | 21 | 29 |
| VC0437 | 7 | 4 | 7 | 11 | 14 | 18 | 25 |
| VC1650 | 9 | 3 | 6 | 8 | 11 | 14 | 19 |
| VCA0171 | 6 | 5 | 9 | 12 | 16 | 21 | 29 |
| VCA0283 | 6 | 5 | 9 | 12 | 16 | 21 | 29 |

The application provides a table with the number of tandem repeats that was found for each locus in the corresponding genomes.

***In silico* PCR MLVA typing.** Additionally to *MLVAtype* and Sanger sequencing methods, an *in silico* PCR method was also assessed. The amplicon sizes were determined from the draft genomes using the vmatchPattern function of the Biostrings R package and subsequently used to derive the number of tandem repeats.

## Results

### GeneScan-derived and Sanger-derived MLVA typing

GeneScan- and Sanger-derived MLVA profiles from DRC and Uganda are reported in Table 3. Considering the high quality of the Sanger sequences, the number of tandem repeats extracted from these sequences were considered as a gold-standard in the current study. Only one mismatch was observed between GeneScan-derived and Sanger-derived MLVA typing.

### WGS-derived MLVA typing

WGS-derived MLVA profiles obtained using the *MLVAtype* algorithm from draft genomes assembled using SPAdes and each value of the k-mer parameter are reported for DRC and Ugandan isolates (Table 4). All estimates of number of tandem repeats appeared to be perfectly

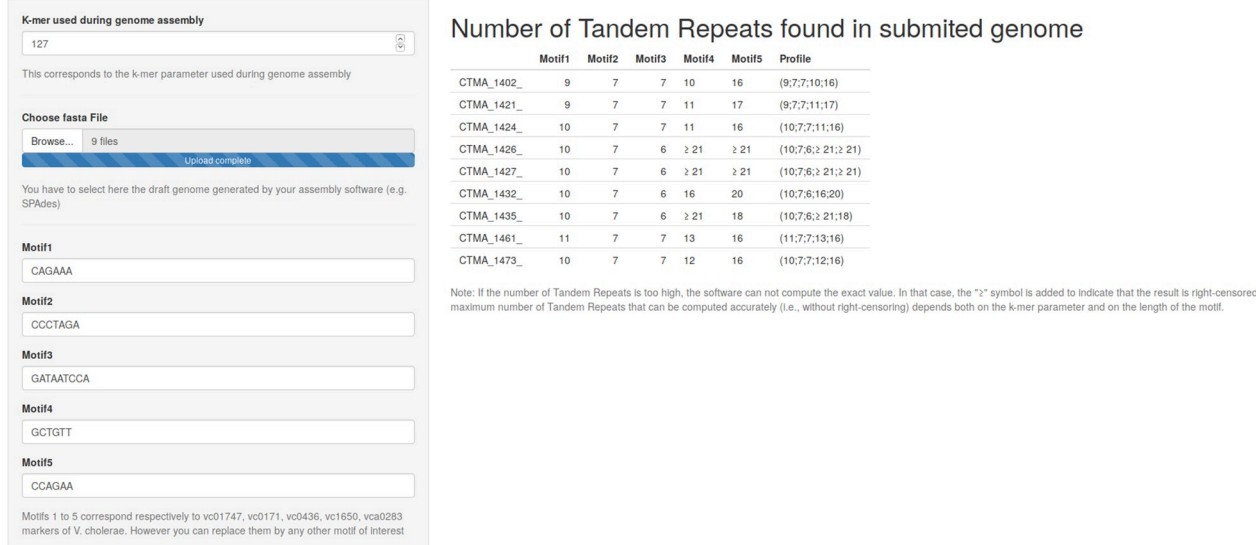

**Fig 2. Screenshot of the *MLVAtype* shiny application.** Nine *V. cholerae* genomes from DRC were uploaded and processed with this application.

**Table 3. MLVA profiles consisting in the number of tandem repetition for each loci (VC0147, VC0437, VC1650, VCA0171, and VCA0283) extracted from GeneScan data and from Sanger sequences.** Mismatch is indicated in bold.

| Country | Isolate | Sanger-derived MLVA profile | GeneScan-derived MLVA profile |
|---|---|---|---|
| Uganda | UG010 | (9;3;7;21;26) | (9,3,7,21,26) |
| Uganda | UG020 | (9;3;7;21;27) | (9,3,7,21,27) |
| Uganda | UG026 | (9;3;7;21;28) | (9,3,7,21,28) |
| Uganda | UG040 | (10;7;7;9;17) | (10,7,7,9,17) |
| Uganda | UG042 | (8;7;7;10;21) | (8,7,7,10,21) |
| Uganda | UG046 | (8;7;7;11;21) | (8,7,7,11,21) |
| Uganda | UG054 | (8;7;7;10;21) | (**5**,7,7,10,21) |
| Uganda | UG060 | (10;7;7;9;17) | (10,7,7,9,17) |
| Uganda | UG071 | (10;7;7;8;18) | (10,7,7,8,18) |
| Uganda | UG086 | (10;7;7;9;18) | (10,7,7,9,18) |

**Table 4. WGS-derived MLVA profiles extracted from DRC and Ugandan genomes assembled with various values of the k-mer parameter and compared to the gold-standard (*i.e.* Sanger-derived MLVA profile).** *: Maximum Number of Tandem Repeats (MNTR) in the corresponding k-mer. n/a: not applicable. Read lengths obtained with DRC and Ugandan isolates were 300 and 150 nt, respectively.

| Country | Isolate | WGS-derived MLVA profile | | | | | Sanger-derived MLVA profile |
|---|---|---|---|---|---|---|---|
| | | k-mer = 55 (9,7,6,9,9)* | k-mer = 77 (12,11,8,12,12)* | k-mer = 99 (16,14,11,16,16)* | k-mer = 127 (21,18,14,21,21)* | k-mer = 175 (29,25,19,29,29)* | |
| DRC | CTMA-1402 | (≥9;≥7;≥6;≥9;≥9) | (9;7;7;10;≥12) | (9;7;7;10;≥16) | (9;7;7;10;16) | (9;7;7;10;16) | (9,7,7,10,16) |
| DRC | CTMA-1421 | (≥9;≥7;≥6;≥9;≥9) | (9;7;7;11;≥12) | (9;7;7;11;≥16) | (9;7;7;11;17) | (9;7;7;11;17) | (9,7,7,11,17) |
| DRC | CTMA-1424 | (≥9;≥7;≥6;≥9;≥9) | (10;7;7;11;≥12) | (10;7;7;11;≥16) | (10;7;7;11;16) | (10;7;7;11;16) | (10,7,7,11,16) |
| DRC | CTMA-1426 | (≥9;≥7;≥6;≥9;≥9) | (10;7;6;≥12;≥12) | (10;7;6;≥16;≥16) | (10;7;6;≥21;≥21) | (10;7;6;24;21) | (10,7,6,24,21) |
| DRC | CTMA-1427 | (≥9;≥7;≥6;≥9;≥9) | (10;7;6;≥12;≥12) | (10;7;6;≥16;≥16) | (10;7;6;≥21;≥21) | (10;7;6;24;21) | (10,7,6,24,21) |
| DRC | CTMA-1432 | (≥9;≥7;≥6;≥9;≥9) | (10;7;6;≥12;≥12) | (10;7;6;≥16;≥16) | (10;7;6;16;20) | (10;7;6;16;20) | (10,7,6,16,20) |
| DRC | CTMA-1435 | (≥9;≥7;≥6;≥9;≥9) | (10;7;6;≥12;≥12) | (10;7;6;≥16;≥16) | (10;7;6;≥21;18) | (10;7;6;23;18) | (10,7,6,23,18) |
| DRC | CTMA-1461 | (≥9;≥7;≥6;≥9;≥9) | (11;7;7;≥12;≥12) | (11;7;7;13;≥16) | (11;7;7;13;16) | (11;7;7;13;16) | (11,7,7,13,16) |
| DRC | CTMA-1473 | (≥9;≥7;≥6;≥9;≥9) | (10;7;7;≥12;≥12) | (10;7;7;12;≥16) | (10;7;7;12;16) | (10;7;7;12;16) | (10,7,7,12,16) |
| Uganda | UG010 | (≥9;3;≥6;≥9;≥9) | (9;3;7;≥12;≥12) | (9;3;7;≥16;≥16) | (9;3;7;≥21;≥21) | n/a | (9,3,7,21,26) |
| Uganda | UG020 | (≥9;3;≥6;≥9;≥9) | (9;3;7;≥12;≥12) | (9;3;7;≥16;≥16) | (9;3;7;≥21;≥21) | n/a | (9,3,7,21,27) |
| Uganda | UG026 | (≥9;3;≥6;≥9;≥9) | (9;3;7;≥12;≥12) | (9;3;7;≥16;≥16) | (9;3;7;≥21;≥21) | n/a | (9,3,7,21,28) |
| Uganda | UG040 | (≥9;≥7;≥6;≥9;≥9) | (10;7;7;9;≥12) | (10;7;7;9;≥16) | (10;7;7;9;17) | n/a | (10,7,7,9,17) |
| Uganda | UG042 | (8;≥7;≥6;≥9;≥9) | (8;7;7;10;≥12) | (8;7;7;10;≥16) | (8;7;7;10;≥21) | n/a | (8,7,7,10,21) |
| Uganda | UG046 | (8;≥7;≥6;≥9;≥9) | (8;7;7;11;≥12) | (8;7;7;11;≥16) | (8;7;7;11;≥21) | n/a | (8,7,7,11,21) |
| Uganda | UG054 | (8;≥7;≥6;≥9;≥9) | (8;7;7;10;≥12) | (8;7;7;10;≥16) | (8;7;7;10;≥21) | n/a | (8,7,7,10,21) |
| Uganda | UG060 | (≥9;≥7;≥6;≥9;≥9) | (10;7;7;9;≥12) | (10;7;7;9;≥16) | (10;7;7;9;17) | n/a | (10,7,7,9,17) |
| Uganda | UG071 | (≥9;≥7;≥6;8;≥9) | (10;7;7;8;≥12) | (10;7;7;8;≥16) | (10;7;7;8;18) | n/a | (10,7,7,8,18) |
| Uganda | UG086 | (≥9;≥7;≥6;≥9;≥9) | (10;7;7;9;≥12) | (10;7;7;9;≥16) | (10;7;7;9;18) | n/a | (10,7,7,9,18) |

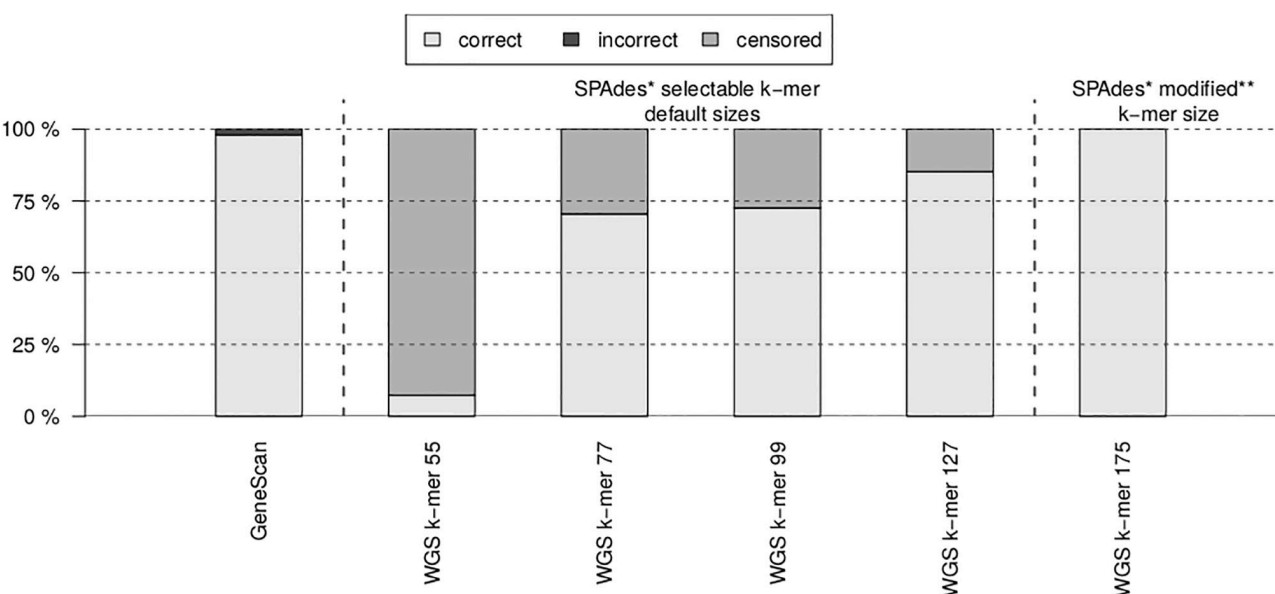

**Fig 3. Percentage of correct, incorrect, and censored estimation of the number of tandem repeats.** Each percentage is produced by either GeneScan (GS) or WGS-based approaches; Sanger sequencing results were used as the reference values. \*: SPAdes v.3.13.0. \*\*: After modification of SPAdes v.3.13.0 source code.

concordant with Sanger-derived typing values. However, the k-mer parameter directly impacts the number of right-censored (*i.e.*, $\geq$) estimations. When a k-mer of 127 was used to assemble reads from DRC and Ugandan isolates, only 5 and 9 right-censored estimations of tandem repeat numbers were produced, respectively. Interestingly, WGS-derived MLVA profiles extracted from 175-mers DRC assemblies were perfectly concordant with Sanger-derived typing results with no censored estimation (Fig 3).

Moreover, the last step of the *MLVAtype* algorithm (*i.e.*, the comparison of the number of tandem repetitions [k] to the MNTR value) was essential. When this step was omitted, several discordances occurred between Sanger- and WGS-derived typing values, especially when using a small k-mer for read assembly (Table 5). When the k-mer size was not specified during genome assembly with SPAdes, resulting MLVA profiles were similar to those obtained with a k-mer of 127 for DRC isolates (300 nt long reads) and a k-mer of 77 for Ugandan isolates (150 nt long reads) (Table 5). When MLVA profiles were derived from draft genomes using the *in silico* PCR approach (Table 6), discordances were observed with Sanger-derived typing results. When using a small k-mer size during genome assembly, forward and reverse primer sequences were often located in different contigs, resulting in an undetermined (U) number of tandem repeats.

It is worth noting that a longer k-mer size also improved the quality of genome assemblies as illustrated by QUAST [11] with lower numbers of contigs and larger N50 values (Fig 4).

## Comparison of analytical reagent costs

Reagents costs for MLVA typing (5 loci) of *V. cholerae* isolates were compared using three different methods (Table 7). The lowest cost was associated with the GeneScan-based method. It is worth nothing that Sanger sequencing, which cost is provided in Table 7, was only used here as a reference method but not intended for routine MLVA typing.

**Table 5. WGS-derived MLVA profiles extracted with *MLVAtype* without taking into account the MNTR value.** Mismatches between WGS- and Sanger-derived values are indicated in bold. n/a: not applicable. Read lengths obtained with DRC and Ugandan isolates were 300 and 150 nt, respectively.

| Country | Isolate | WGS-derived MLVA profile | | | | | | Sanger-derived MLVA profile |
|---|---|---|---|---|---|---|---|---|
| | | k-mer = 55 | k-mer = 77 | k-mer = 99 | k-mer = 127 | k-mer = 175 | k-mer unsp. | |
| DRC | CTMA-1402 | (9;**7**;**6**;**9**;**15**) | (9;7;7;10;**13**) | (9;7;7;10;16) | (9;7;7;10;16) | (9;7;7;10;16) | (9;7;7;10;16) | (9,7,7,10,16) |
| DRC | CTMA-1421 | (9;**7**;**6**;**9**;**10**) | (9;7;7;11;**14**) | (9;7;7;11;17) | (9;7;7;11;17) | (9;7;7;11;17) | (9;7;7;11;17) | (9,7,7,11,17) |
| DRC | CTMA-1424 | (**9**;**7**;**6**;**9**;**9**) | (10;7;7;11;**13**) | (10;7;7;11;16) | (10;7;7;11;16) | (10;7;7;11;16) | (10;7;7;11;16) | (10,7,7,11,16) |
| DRC | CTMA-1426 | (**9**;7;6;**10**;**13**) | (10;7;6;**13**;**13**) | (10;7;6;**17**;**17**) | (10;7;6;**21**;21) | (10;7;6;24;21) | (10;7;6;**21**;21) | (10,7,6,24,21) |
| DRC | CTMA-1427 | (**9**;7;6;**10**;**13**) | (10;7;6;**13**;**14**) | (10;7;6;**17**;**17**) | (10;7;6;**21**;21) | (10;7;6;24;21) | (10;7;6;**21**;21) | (10,7,6,24,21) |
| DRC | CTMA-1432 | (**9**;7;6;**9**;**9**) | (10;7;6;**13**;**13**) | (10;7;6;16;**17**) | (10;7;6;16;20) | (10;7;6;16;20) | (10;7;6;16;20) | (10,7,6,16,20) |
| DRC | CTMA-1435 | (**9**;7;6;**10**;**10**) | (10;7;6;**13**;**13**) | (10;7;6;**16**;**17**) | (10;7;6;**21**;18) | (10;7;6;23;18) | (10;7;6;**21**;18) | (10,7,6,23,18) |
| DRC | CTMA-1461 | (**9**;7;6;**10**;**10**) | (11;7;7;**12**;**14**) | (11;7;7;13;16) | (11;7;7;13;16) | (11;7;7;13;16) | (11;7;7;13;16) | (11,7,7,13,16) |
| DRC | CTMA-1473 | (**9**;7;6;**10**;**15**) | (10;7;7;12;**13**) | (10;7;7;12;16) | (10;7;7;12;16) | (10;7;7;12;16) | (10;7;7;12;16) | (10,7,7,12,16) |
| Uganda | UG010 | (9;3;**6**;**10**;**9**) | (9;3;7;**13**;**13**) | (9;3;7;**16**;**16**) | (9;3;7;21;**21**) | n/a | (9;3;7;**13**;**13**) | (9;3;7;21;26) |
| Uganda | UG020 | (9;3;**6**;**10**;**9**) | (9;3;7;**13**;**13**) | (9;3;7;**16**;**16**) | (9;3;7;21;**21**) | n/a | (9;3;7;**13**;**13**) | (9;3;7;21;27) |
| Uganda | UG026 | (9;3;**6**;**10**;**9**) | (9;3;7;**13**;**13**) | (9;3;7;**16**;**16**) | (9;3;7;21;**21**) | n/a | (9;3;7;**13**;**13**) | (9;3;7;21;28) |
| Uganda | UG040 | (**9**;7;6;9;**9**) | (10;7;7;9;**13**) | (10;7;7;9;**16**) | (10;7;7;9;17) | n/a | (10;7;7;9;**13**) | (10;7;7;9;17) |
| Uganda | UG042 | (8;7;**6**;**9**;**9**) | (8;7;7;10;**13**) | (8;7;7;10;**16**) | (8;7;7;10;21) | n/a | (8;7;7;10;**13**) | (8;7;7;10;21) |
| Uganda | UG046 | (8;7;**6**;**9**;**9**) | (8;7;7;11;**13**) | (8;7;7;11;**16**) | (8;7;7;11;21) | n/a | (8;7;7;11;**13**) | (8;7;7;11;21) |
| Uganda | UG054 | (8;7;**6**;**9**;**9**) | (8;7;7;10;**13**) | (8;7;7;10;**16**) | (8;7;7;10;21) | n/a | (8;7;7;10;**13**) | (8;7;7;10;21) |
| Uganda | UG060 | (**9**;7;6;9;**9**) | (10;7;7;9;**13**) | (10;7;7;9;**16**) | (10;7;7;9;17) | n/a | (10;7;7;9;**13**) | (10;7;7;9;17) |
| Uganda | UG071 | (**9**;7;6;8;**20**) | (10;7;7;8;**13**) | (10;7;7;8;**16**) | (10;7;7;8;18) | n/a | (10;7;7;8;**13**) | (10;7;7;8;18) |
| Uganda | UG086 | (**9**;7;6;9;**20**) | (10;7;7;9;**13**) | (10;7;7;9;16) | (10;7;7;9;18) | n/a | (10;7;7;9;**13**) | (10;7;7;9;18) |

**Table 6. WGS-derived MLVA profiles extracted using an *in silico* PCR approach.** Mismatches between WGS- and Sanger-derived values are indicated in bold. U: undetermined. n/a: not applicable. Read lengths obtained with DRC and Ugandan isolates were 300 and 150 nt, respectively.

| Country | Isolate | WGS-derived MLVA profile | | | | | | Sanger-derived MLVA profile |
|---|---|---|---|---|---|---|---|---|
| | | k-mer = 55 | k-mer = 77 | k-mer = 99 | k-mer = 127 | k-mer = 175 | k-mer unsp. | |
| DRC | CTMA-1402 | (9;**7**;**6**;**9**;U) | (9;7;7;10;U) | (9;7;7;10;16) | (9;7;7;10;16) | (9;7;7;10;16) | (9;7;7;10;16) | (9,7,7,10,16) |
| DRC | CTMA-1421 | (9;**7**;**6**;**9**;U) | (9;7;7;11;U) | (9;7;7;11;17) | (9;7;7;11;17) | (9;7;7;11;17) | (9;7;7;11;17) | (9,7,7,11,17) |
| DRC | CTMA-1424 | (**9**;**7**;**6**;**9**;U) | (10;7;7;11;**13**) | (10;7;7;11;16) | (10;7;7;11;16) | (10;7;7;11;16) | (10;7;7;11;16) | (10,7,7,11,16) |
| DRC | CTMA-1426 | (**9**;7;6;U;U) | (10;7;6;U;U) | (10;7;6;U;**17**) | (10;7;6;**21**;21) | (10;7;6;24;21) | (10;7;6;**21**;21) | (10,7,6,24,21) |
| DRC | CTMA-1427 | (**9**;7;6;U;U) | (10;7;6;U;U) | (10;7;6;U;**17**) | (10;7;6;**21**;21) | (10;7;6;24;21) | (10;7;6;**21**;21) | (10,7,6,24,21) |
| DRC | CTMA-1432 | (**9**;7;6;U;U) | (10;7;6;U;**13**) | (10;7;6;16;**17**) | (10;7;6;16;20) | (10;7;6;16;20) | (10;7;6;16;20) | (10,7,6,16,20) |
| DRC | CTMA-1435 | (**9**;7;6;U;U) | (10;7;6;U;**13**) | (10;7;6;U;**17**) | (10;7;6;**21**;18) | (10;7;6;23;18) | (10;7;6;**21**;18) | (10,7,6,23,18) |
| DRC | CTMA-1461 | (**9**;7;6;U;U) | (11;7;7;**12**;U) | (11;7;7;13;16) | (11;7;7;13;16) | (11;7;7;13;16) | (11;7;7;13;16) | (11,7,7,13,16) |
| DRC | CTMA-1473 | (**9**;7;6;U;U) | (10;7;7;12;U) | (10;7;7;12;16) | (10;7;7;12;16) | (10;7;7;12;16) | (10;7;7;12;16) | (10,7,7,12,16) |
| Uganda | UG010 | (9;3;**6**;U;U) | (9;3;7;U;U) | (9;3;7;U;U) | (9;3;7;21;U) | n/a | (9;3;7;**13**;**13**) | (9;3;7;21;26) |
| Uganda | UG020 | (9;3;**6**;U;U) | (9;3;7;U;U) | (9;3;7;**16**;U) | (9;3;7;21;U) | n/a | (9;3;7;**13**;**13**) | (9;3;7;21;27) |
| Uganda | UG026 | (9;3;**6**;U;U) | (9;3;7;U;U) | (9;3;7;**16**;U) | (9;3;7;21;U) | n/a | (9;3;7;**13**;**13**) | (9;3;7;21;28) |
| Uganda | UG040 | (**9**;7;6;9;U) | (10;7;7;9;**13**) | (10;7;7;9;**17**) | (10;7;7;9;17) | n/a | (10;7;7;9;**13**) | (10;7;7;9;17) |
| Uganda | UG042 | (8;7;**6**;**9**;U) | (8;7;7;10;U) | (8;7;7;10;**17**) | (8;7;7;10;21) | n/a | (8;7;7;10;**13**) | (8;7;7;10;21) |
| Uganda | UG046 | (8;7;**6**;**9**;U) | (8;7;7;11;U) | (8;7;7;11;**17**) | (8;7;7;11;21) | n/a | (8;7;7;11;**13**) | (8;7;7;11;21) |
| Uganda | UG054 | (8;7;**6**;**9**;U) | (8;7;7;10;U) | (8;7;7;10;**17**) | (8;7;7;10;21) | n/a | (8;7;7;10;**13**) | (8;7;7;10;21) |
| Uganda | UG060 | (**9**;7;6;9;U) | (10;7;7;9;U) | (10;7;7;9;**17**) | (10;7;7;9;17) | n/a | (10;7;7;9;**13**) | (10;7;7;9;17) |
| Uganda | UG071 | (**9**;7;6;8;U) | (10;7;7;8;**13**) | (10;7;7;8;**17**) | (10;7;7;8;18) | n/a | (10;7;7;8;**13**) | (10;7;7;8;18) |
| Uganda | UG086 | (**9**;7;6;9;U) | (10;7;7;9;**13**) | (10;7;7;9;**17**) | (10;7;7;9;18) | n/a | (10;7;7;9;**13**) | (10;7;7;9;18) |

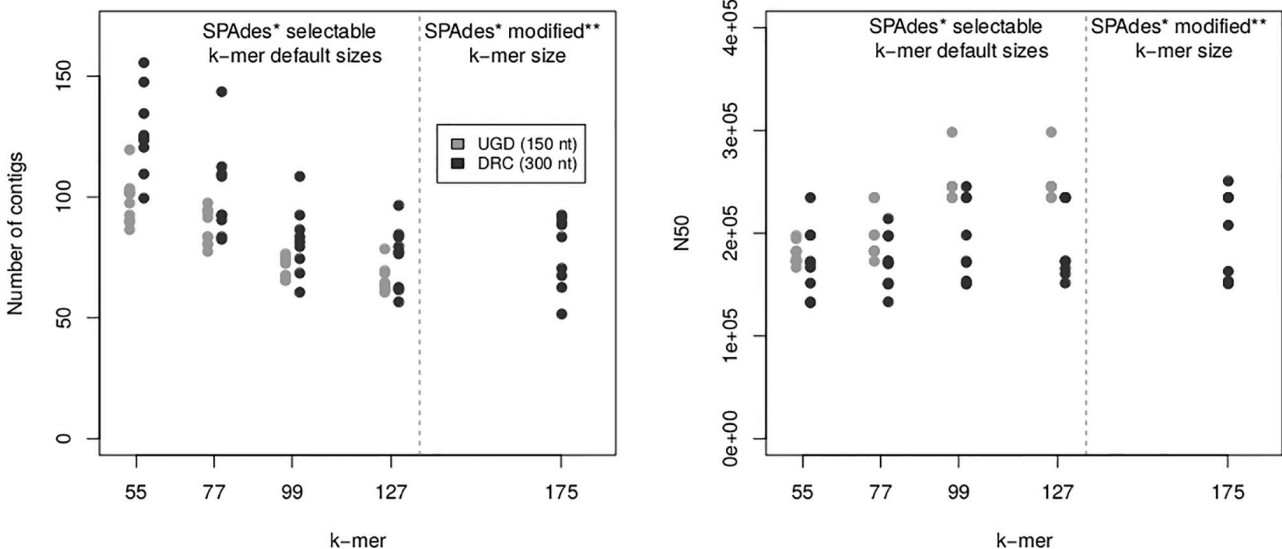

**Fig 4. Quality metrics of genome assemblies.** Number of contigs and N50 metrics, as reported by QUAST for genome assemblies from DRC and Ugandan (UGD) isolates and various selectable default and modified k-mer sizes. *: SPAdes v.3.13.0. **: After modification of the SPAdes v.3.13.0 source code.

## Discussion

The rationale behind this study is the dramatic increase in the rate and amount of sequencing and a continuous decrease in sequencing costs. The efficiency in microbial identification and subtyping allowed by WGS characterization makes it now a reference method with a clear potential to replace traditional typing methods. Anyhow, this type of analysis introduces new challenges, *i.e.* data storage, computing power, bioinformatics expertise and results returned in real-time, all potentially related to high costs. Undoubtedly, the WGS analysis is not yet affordable in all institutions or networks in charge of pathogen surveillance.

Accordingly, we present here a new application enabling users to extract *V. cholerae* MLVA profiles from WGS data while preserving their backward compatibility with conventional MLVA typing. To validate this application, MLVA profiles generated by *MLVAType* were compared to Sanger-derived MLVA profiles on nineteen *V. cholerae* isolates among which 9 from DRC and 10 from Uganda, and the respective costs were assessed. In addition, Sanger-derived MLVA profiles of the 9 isolates from Uganda were compared to GeneScan-derived MLVA profiles.

There was only one mismatch found between the GeneScan-derived and Sanger reference method which could not be explained but does not seem unusual [13]. However, it should be

**Table 7. Cost of MLVA targeting five loci.** *: Optimal cost if the five loci are amplified in a single PCR (theoretical cost analysis). **: Reported cost of a typing method based on a duplex and a triplex PCR followed by two fragment analysis runs. ***: Cost of commercial sequencing per sample in 2019. **** Reagent and consumable cost of Illumina sequencing per sample [12].

| | Cost USD | | | |
|---|---|---|---|---|
| | **Sanger-based (5 loci)** | **GeneScan-based (5 loci) optimal cost*** | **GeneScan-based (5 loci) published protocol**** | **WGS-based** |
| PCR | 40 (5 x 8) | 8 | 16 (2 x 8) | - |
| Analytical cost | 35 (5 x 7) | 3.4 | 6.8 (2 x 3.4) | 99–150 |
| Total | **75** | **11.4** | 22.8 | **99***–150***** |

noted that a modification of either the primer or the formula proposed by Kendall *et al.* [10], which determines *V. cholerae* VNTR repeat numbers, was applied in order to decrease the number of mismatches (Table 1), especially those observed with VC0437 and VCA0283.

Interestingly, there was no mismatch between WGS- and Sanger-derived MLVA profiles. However, WGS-derived profiles were affected by censored estimations whose proportion varied according to the k-mer size used during genome assembly with SPAdes: the larger, the k-mer size, the better the accuracy of WGS-derived MLVA profiles (Fig 3). Accordingly, it is recommended to use the largest possible k-mer size to assemble the reads into contigs before determining the number of tandem repeats, but considering that this maximum value is inherent to the WGS read length, as illustrated with the current application, using WGS data from *V. cholerae*. While using a k-mer size of 175 was inapplicable with read length shorter than 175, as for Ugandan isolates, it did not produce censored data with 300 nt reads from DRC isolates.

While WGS read lengths and qualities increase, one may expect to increase k-mer sizes to perform *in silico* MLVA typing using this *MLVAType* application whenever data comparison with MLVA database is needed. As previously reported, the length of repeat motifs should not exceed 174 nucleotides for *V. cholerae*, corresponding to 29 repetitions of a 6 nt motif [14]. Accordingly, the longer k-mer size (*i.e.* 175) proved to generate a correct MLVA profile with no censored data.

Importantly, the *MLVAType* algorithm was developed to extract MLVA profiles of *V. cholerae* isolates which are characterized by (i) a perfect repeat array, (ii) locus-specific VNTRs (*i.e.* there is no repeat unit in other loci), and (iii) the absence of indels in the flanking region. Whereas the *in silico* PCR approach was found inefficient for MLVA typing of *V. cholerae* (Table 6), MLVAType could not be used for species such as *Brucella* or *Salmonella*. This makes MLVAType a complementary tool to the existing *in silico* PCR approach when the MLVA profiles need to be retrieved from WGS data.

## Conclusion

In conclusion, the *MLVAType* shiny application proved to extract reliably MLVA profiles of *V. cholerae* isolates from WGS data. Considering the wide *in silico* exploitation of WGS data, our perspective will then be to combine the extracted information related both to VNTRs and Single Nucleotide Variants (SNVs), and to calculate a single genetic relatedness index. This should further extend our understanding of the genetic relatedness of *V. cholerae* isolates while giving us better insight into how the VNTRs evolve over time.

## Acknowledgments

The authors are grateful to the following institutions for exchange of information and expertise provided during the course of this work: Uganda Ministry of Health, DR Congo Ministry of Health, Makerere University, John Hopkins Bromberg School of Public Health, Maryland University of Medicine, and Defense Laboratory Department (DLD), Belgium. The authors would like in a special way thank the following persons for their wonderful contribution and guidance; Professor David A. Sack, Professor Christopher G. Orach, Mr. Atek. Kagirita, Mr. Mathieu Almeida, Dr. Amanda K. Debes, M/s Shan Li, Mr. JB.Voeglein, and Dr Prudence Mitangala.

## Author Contributions

**Conceptualization:** Jérôme Ambroise, Léonid M. Irenge.

**Formal analysis:** Jérôme Ambroise, Jean-François Durant.

**Funding acquisition:** Jean-Luc Gala.

**Investigation:** Jérôme Ambroise, Jean-François Durant, Bertrand Bearzatto, O. Colin Stine, Jean-Luc Gala.

**Methodology:** Jérôme Ambroise, Léonid M. Irenge, O. Colin Stine, Jean-Luc Gala.

**Resources:** Jean-Luc Gala.

**Software:** Jérôme Ambroise.

**Supervision:** O. Colin Stine, Jean-Luc Gala.

**Validation:** Jérôme Ambroise, Godfrey Bwire.

**Visualization:** Jérôme Ambroise.

**Writing – original draft:** Jérôme Ambroise, O. Colin Stine, Jean-Luc Gala.

**Writing – review & editing:** Jérôme Ambroise, Bertrand Bearzatto, Godfrey Bwire, O. Colin Stine, Jean-Luc Gala.

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
