## [Decision Letter · Decision Letter 0]

12 Aug 2019

PONE-D-19-20834

Backward compatibility of whole genome sequencing data with MLVA typing using a new MLVAtype shiny application: the example of Vibrio cholerae

PLOS ONE

Dear Dr Ambroise,

Thank you for submitting your manuscript to PLOS ONE. After careful consideration, we feel that it has merit but does not fully meet PLOS ONE’s publication criteria as it currently stands. Therefore, we invite you to submit a revised version of the manuscript that addresses the points raised during the review process.

Please consider the comments of the reviewer to improve the manuscript.

We would appreciate receiving your revised manuscript by Sep 26 2019 11:59PM. To enhance the reproducibility of your results, we recommend that if applicable you deposit your laboratory protocols in protocols.io, where a protocol can be assigned its own identifier (DOI) such that it can be cited independently in the future. For instructions see: http://journals.plos.org/plosone/s/submission-guidelines#loc-laboratory-protocols

We look forward to receiving your revised manuscript.

Kind regards,

Axel Cloeckaert

Academic Editor

PLOS ONE

Journal Requirements:

3. In your Methods section, please clarify whether the isolates were obtained from a collection, a company, or from another third party source

Reviewers' comments:

Reviewer's Responses to Questions

**Comments to the Author**

1. Is the manuscript technically sound, and do the data support the conclusions?

Reviewer #1: No

2. Has the statistical analysis been performed appropriately and rigorously? 

Reviewer #1: No

3. Have the authors made all data underlying the findings in their manuscript fully available?

Reviewer #1: No

4. Is the manuscript presented in an intelligible fashion and written in standard English?

Reviewer #1: Yes

5. Review Comments to the Author

Reviewer #1: Ambroise et al describe a software (MLVAtype) which allows to deduce MLVA genotypes from assembled Whole Genome Sequence (WGS) data. The input is a draft assembly, the size of the k-mer used for the assembly and the sequence of the repeated motif. MLVAtype will search for the largest occurrence of repeated motifs. Then MLVAtype will compare the length of this largest occurrence and will consider it as a valid VNTR allele if it is smaller than the k-mer value used for the assembly.

To evaluate MLVAtype, the authors used 19 V. cholera isolates and an MLVA assay comprising five loci with a 6 bp (3 loci), 7 bp or 9 bp repeat unit. They sequenced the PCR amplification products by Sanger sequencing to constitute the reference data. They then compare in vitro and WGS-derived MLVA typing data with the reference data.

Nine isolates were typed in vitro using the Agilent 2100 Bioanalyzer whereas ten isolates were typed in vitro using the higher resolution capillary electrophoresis system GenScan. 300 bp long WGS data was produced for the nine isolates versus 150 bp for the ten isolates. The 300 bp long reads allow to deduce a full and correct MLVA genotype whereas the 150 bp reads provide partial data. 90% of in vitro MLVA profiles are incorrect.

There are a number of issues associated with the software design, in vitro typing, report organization.

-in some species in which MLVA is used, some VNTRs occur in families, i.e. different VNTR loci share the same repeat unit. How will MLVAtype behave in such a case? It seems that all loci sharing an identical repeat motif will be (incorrectly) assigned the largest allele size.

-conversely, tandem repeats are often not perfect. How will MLVAtype behave in such a case?

-what is the rational for not taking into account contigs smaller than 2000 bp, especially given that in the present situation (Vibrio cholera VNTRs), the VNTR loci are shorter than 200 bp?

-Page 2 third $: the authors do not mention other published approaches for in silico MLVA typing. For instance Vergnaud et al. Frontiers Microbiology 2018 previously explored the possibility to deduce MLVA from WGS data was evaluated for Brucella using an in silico PCR approach.

-Page 6, Table 4: please also provide the initial estimates before applying the “right-censorchip”. The authors seem to implicitly assume that Spades will never correctly reconstruct tandem repeat arrays longer than the k-mer size, but do not provide the data to show that.

-by default, Spades will explore multiple k-mer values (rather than a specified value). It would seem useful to provide in Table 4 the results obtained in these conditions.

-if indeed Spades will not reconstruct tandem repeats longer than k-mer size, then why try to assemble reads when the read length is longer than available k-mer sizes (instead of recovering the reads of interest using tools such as BBduk)?

-the in vitro MLVA assay does not seem to be correctly working yet, because the Bioanalyzer does not have a sufficient amplicon sizing resolution and/or because the allele calling (conversion from size estimate to repeat copy number) is not optimized, as indeed suggested by the authors in the first paragraph page 8. The error rates reported in Table 3 for the Bioanalyzer (above 50%!) and for GenScan (7 errors in 10 strains for the second locus) are not acceptable. What is the interest of backward compatibility of WGS with in vitro MLVA typing if the in vitro data is so bad, i.e. if MLVA does not seem applicable here? I believe that the Bioanalyzer (in)capacity to discriminate VNTR alleles with repeat units smaller than 8 bp has already been discussed in the literature, see for instance De Santis et al., BMC microbiology 2011. Regarding the GenScan errors, this is probably due to incorrect allele calling, resulting from slightly inexact size measurement by the capillary equipment (literature available, see for instance Hyytia-Trees et al., Foodborne Pathog Dis 2010). Once correctly set-up, there should be no errors at least when using the GenScan.

Table 5: include the cost estimate for the GeneScan method (where the assay can be run in a single multiplex PCR).

The estimated cost for PCR (10 € per PCR for reagent costs) seems a bit high.

Please clarify the indicated WGS cost: does this cover the making of the sequencing library? What read length?

More generally, Table 5 and the associated paragraph are poorly informative, it would be useful to try to estimate the overall cost, based upon commercial services prices.

Page 7, paragraph on “Theoretical feasibility …” is not informative as is. Should either be developed, by being more specific, or deleted.

Page 8, second paragraph, “the larger, the k-mer size, the better the accuracy of WGS-derived MLVA profiles.” The authors need to show the data (see previous remark on Table 4). Also please explain the reason for limiting the k-mer size to 175

Page 8, last paragraph before conclusion: the theoretical evaluation has not been done appropriately. The postulated 6 nt motif is not applicable to any of the MLVA assays commonly used in the given list of genus/species. Rather the design of the MLVAtype software indicates that it will have a very limited range. Indeed the MLVAtype web page at https://ucl-irec-ctma.shinyapps.io/NGS-MLVA-TYPING/ appears tailored for V. cholerae (and perhaps MLVA assays with up to five VNTR loci, and short and perfect repeat arrays). The article might be more convincing if focused on V. cholerae and its 10000 publicly available sequence reads archives.

-details,

Page 1 last $, Mycobacterium, Streptococcus etc are not species but genus. Please be more specific.

Page 2, first paragraph “Mounting evidence …” obviously and by definition Whole Genome Sequencing can only be better than the previous methods, no need to refer to “Mounting evidence”. The issue is cost, as detailed in the second paragraph.

6. PLOS authors have the option to publish the peer review history of their article (what does this mean?). If published, this will include your full peer review and any attached files.

Reviewer #1: No

---

## [Author Response · Author response to Decision Letter 0]

23 Sep 2019

We thank the Editor and the Reviewer for their careful reading of our manuscript. In revising our paper, we carefully followed the editor’s direction, and replied to the point by point questions of the reviewer. 

We are confident that the answers provided and the corresponding modifications in the revised version will now meet the Editor and the Reviewer’s expectations. 

We thank you in advance for your editorial work. 

The authors 

Reviewer #1: Ambroise et al describe a software (MLVAtype) which allows to deduce MLVA genotypes from assembled Whole Genome Sequence (WGS) data. The input is a draft assembly, the size of the k-mer used for the assembly and the sequence of the repeated motif. MLVAtype will search for the largest occurrence of repeated motifs. Then MLVAtype will compare the length of this largest occurrence and will consider it as a valid VNTR allele if it is smaller than the k-mer value used for the assembly.

To evaluate MLVAtype, the authors used 19 V. cholera isolates and an MLVA assay comprising five loci with a 6 bp (3 loci), 7 bp or 9 bp repeat unit. They sequenced the PCR amplification products by Sanger sequencing to constitute the reference data. They then compare in vitro and WGS-derived MLVA typing data with the reference data.

Nine isolates were typed in vitro using the Agilent 2100 Bioanalyzer whereas ten isolates were typed in vitro using the higher resolution capillary electrophoresis system GenScan. 300 bp long WGS data was produced for the nine isolates versus 150 bp for the ten isolates. The 300 bp long reads allow to deduce a full and correct MLVA genotype whereas the 150 bp reads provide partial data. 90% of in vitro MLVA profiles are incorrect.

There are a number of issues associated with the software design, in vitro typing, report organization.

Question: 

-in some species in which MLVA is used, some VNTRs occur in families, i.e. different VNTR loci share the same repeat unit. How will MLVAtype behave in such a case? It seems that all loci sharing an identical repeat motif will be (incorrectly) assigned the largest allele size.

Answer:

All loci sharing an identical repeat motif will indeed be incorrectly assigned the largest allele size. For such species, an in silico PCR approach would be more appropriate. According to your recommendation (see your other later comment), we decided to focus the paper on V. cholerae where each repeat unit is only found in one locus. 

Question:

-conversely, tandem repeats are often not perfect. How will MLVAtype behave in such a case?

Answer:

For such case, an in silico PCR approach would indeed be more appropriate. According to your recommendation (see your comment later), we have now restricted the focus of our application on V. cholerae where tandem repeats are perfect. It is worth noting that the problem of right-censoring applies only to a perfect repetition of the same motif. This also explains why right-censored data were not observed in the study of Vergnaud et al. Frontiers Microbiology 2018. 

Question:

-what is the rational for not taking into account contigs smaller than 2000 bp, especially given that in the present situation (Vibrio cholera VNTRs), the VNTR loci are shorter than 200 bp?

Answer:

The 2000 bp threshold was modified to 1000 bp in order to be in line with literature. Small contigs are actually excluded because they are often non-informative [1, 2, 3 , 4] and associated with a low coverage.

Question:

-Page 2 third $: the authors do not mention other published approaches for in silico MLVA typing. For instance Vergnaud et al. Frontiers Microbiology 2018 previously explored the possibility to deduce MLVA from WGS data was evaluated for Brucella using an in silico PCR approach.

Answer:

We fully agree with the reviewer and thank him for his suggestion. This reference is now included in the reference list of the amended version of the paper.

Question:

-Page 6, Table 4: please also provide the initial estimates before applying the “right-censorchip”. The authors seem to implicitly assume that Spades will never correctly reconstruct tandem repeat arrays longer than the k-mer size, but do not provide the data to show that.

Answer:

We agree with the comment. In the amended version of the paper, this table is now provided as ‘supplementary file 1’ and commented in the “Results” section.

Question:

-by default, Spades will explore multiple k-mer values (rather than a specified value). It would seem useful to provide in Table 4 the results obtained in these conditions.

Answer:

We agree with the comment. In the amended version of the paper, these results are now provided in the ‘supplementary file 1’ and commented in the “Results” section.

Question:

-if indeed Spades will not reconstruct tandem repeats longer than k-mer size, then why try to assemble reads when the read length is longer than available k-mer sizes (instead of recovering the reads of interest using tools such as Bbduk)?

Answer:

One advantage of the current application is that you can run the MLVA typing on a shiny application very quickly and easily. This process (including data transfer and analyses) would be much longer if the MLVA profiles were directly typed from the reads. Therefore, deriving MLVA profiles directly from the reads was not tested in our study. As the current results showed no discrepancy between WGS data (after assembly) and Sanger sequencing data (gold standard), we focused the paper on deriving MLVA profiles from the assembly. 

Question:

-the in vitro MLVA assay does not seem to be correctly working yet, because the Bioanalyzer does not have a sufficient amplicon sizing resolution and/or because the allele calling (conversion from size estimate to repeat copy number) is not optimized, as indeed suggested by the authors in the first paragraph page 8. The error rates reported in Table 3 for the Bioanalyzer (above 50%!) and for GenScan (7 errors in 10 strains for the second locus) are not acceptable. What is the interest of backward compatibility of WGS with in vitro MLVA typing if the in vitro data is so bad, i.e. if MLVA does not seem applicable here? I believe that the Bioanalyzer (in)capacity to discriminate VNTR alleles with repeat units smaller than 8 bp has already been discussed in the literature, see for instance De Santis et al., BMC microbiology 2011. Regarding the GenScan errors, this is probably due to incorrect allele calling, resulting from slightly inexact size measurement by the capillary equipment (literature available, see for instance Hyytia-Trees et al., Foodborne Pathog Dis 2010). Once correctly set-up, there should be no errors at least when using the GenScan.

Answer:

The objective of the paper was not to optimize Bioanalyzer-derived MLVA typing (albeit this can of course be done as previously reported by Lista et a. in the literature). Accordingly and to avoid any ambiguity about the imperfect Bioanalyzer results presented in the first version of the paper, this part was removed in the amended version.

Likewise, and as discussed supra, the purpose was to compare NGs results with existing published results. We did not really questioned the reliability of these published GenScan results. However, following the comment of the reviewers, we addressed this question to our foreign collaborators and they decide to review the initial GenScan-based MLVA profiles; a senior technician ran them again: all but one - unexplained - mismatches were corrected! 

These errors in our first submission and the reviewer’s comment underpin therefore the value of the work of Hyytia-Trees et al concluding that “proper training and experience is necessary to collect accurate information when using the GeneScan methodology”. 

Albeit not 100% concordant, this concordance between Sanger- and GenScan-derived MLVA profiles is substantially improved in the amended version of the manuscript (Table 3, Figure 3).

Question:

Table 5: include the cost estimate for the GeneScan method (where the assay can be run in a single multiplex PCR).

The estimated cost for PCR (10 € per PCR for reagent costs) seems a bit high.

Please clarify the indicated WGS cost: does this cover the making of the sequencing library? What read length?

More generally, Table 5 and the associated paragraph are poorly informative, it would be useful to try to estimate the overall cost, based upon commercial services prices.

Answer:

The cost of the Bioanalyzer method was replaced by the cost of the GenScan method.

In addition, PCR costs were updated and a new reference was added to clarify the cost of WGS analysis. 

Question:

Page 7, paragraph on “Theoretical feasibility …” is not informative as is. Should either be developed, by being more specific, or deleted.

Answer:

This paragraph was deleted, accordingly.

Question:

Page 8, second paragraph, “the larger, the k-mer size, the better the accuracy of WGS-derived MLVA profiles.” The authors need to show the data (see previous remark on Table 4). Also please explain the reason for limiting the k-mer size to 175.

Answer:

Page 8, second paragraph is based on Figure 3. This is specified in the amended version of the paper. 

The reason for limiting the k-mer to 175 is justified as follows: 

“As previously reported, the length of repeat motifs should not exceed 174 nucleotides for V. cholerae, corresponding to 29 repetitions of a 6 nt motif [13]. Accordingly, the longer k-mer size (i.e. 175) proved to generate a correct MLVA profile with no censored data.”

Question:

Page 8, last paragraph before conclusion: the theoretical evaluation has not been done appropriately. The postulated 6 nt motif is not applicable to any of the MLVA assays commonly used in the given list of genus/species. Rather the design of the MLVAtype software indicates that it will have a very limited range. Indeed the MLVAtype web page at https://ucl-irec-ctma.shinyapps.io/NGS-MLVA-TYPING/ appears tailored for V. cholerae (and perhaps MLVA assays with up to five VNTR loci, and short and perfect repeat arrays). The article might be more convincing if focused on V. cholerae and its 10000 publicly available sequence reads archives.

Answer:

We fully agree with the reviewer’s suggestion. The updated version of the paper focuses now only on V. cholerae MLVA application. 

Question:

-details,

Page 1 last $, Mycobacterium, Streptococcus etc are not species but genus. Please be more specific.

Answer

Considering that the new version of the paper focuses on V. cholerae, and that the paragraph related to the theoretical feasibility has been removed in the new version of the paper, this sentence was removed.

Question:

Page 2, first paragraph “Mounting evidence …” obviously and by definition Whole Genome Sequencing can only be better than the previous methods, no need to refer to “Mounting evidence”. The issue is cost, as detailed in the second paragraph.

Answer

This sentence has been removed, as required.

References:

1: Gurevich, Alexey, et al. "QUAST: quality assessment tool for genome assemblies." Bioinformatics 29.8 (2013): 1072-1075.

2: Bultman, Katherine M., et al. "Draft Genome Sequences of Type VI Secretion System-Encoding Vibrio fischeri Strains FQ-A001 and ES401." Microbiology resource announcements 8.20 (2019): e00385-19.

3: Rozanov, Aleksey S., et al. "Metagenome-Assembled Genome Sequence of Phormidium sp. Strain SL48-SHIP, Isolated from the Microbial Mat of Salt Lake Number 48 (Novosibirsk Region, Russia)." Microbiology resource announcements 8.31 (2019): e00651-19.

4: Parks, Dylan, et al. "Genome Sequence of Bacillus subtilis natto VK161, a Novel Strain That Produces Vitamin K2." Microbiology resource announcements 8.35 (2019): e00444-19.

---

## [Decision Letter · Decision Letter 1]

8 Oct 2019

PONE-D-19-20834R1

Backward compatibility of whole genome sequencing data with MLVA typing using a new MLVAtype shiny application for Vibrio cholerae

PLOS ONE

Dear Dr Ambroise,

Thank you for submitting your manuscript to PLOS ONE. After careful consideration, we feel that it has merit but does not fully meet PLOS ONE’s publication criteria as it currently stands. Therefore, we invite you to submit a revised version of the manuscript that addresses the points raised during the review process.

Please consider the comments of the reviewer to improve the manuscript.

We would appreciate receiving your revised manuscript by Nov 22 2019 11:59PM. To enhance the reproducibility of your results, we recommend that if applicable you deposit your laboratory protocols in protocols.io, where a protocol can be assigned its own identifier (DOI) such that it can be cited independently in the future. For instructions see: http://journals.plos.org/plosone/s/submission-guidelines#loc-laboratory-protocols

We look forward to receiving your revised manuscript.

Kind regards,

Axel Cloeckaert

Academic Editor

PLOS ONE

Reviewers' comments:

Reviewer's Responses to Questions

**Comments to the Author**

1. If the authors have adequately addressed your comments raised in a previous round of review and you feel that this manuscript is now acceptable for publication, you may indicate that here to bypass the “Comments to the Author” section, enter your conflict of interest statement in the “Confidential to Editor” section, and submit your "Accept" recommendation.

Reviewer #1: (No Response)

2. Is the manuscript technically sound, and do the data support the conclusions?

Reviewer #1: Partly

3. Has the statistical analysis been performed appropriately and rigorously? 

Reviewer #1: N/A

4. Have the authors made all data underlying the findings in their manuscript fully available?

Reviewer #1: Yes

5. Is the manuscript presented in an intelligible fashion and written in standard English?

Reviewer #1: Yes

6. Review Comments to the Author

Reviewer #1: The authors have significantly clarified and improved their report. A few points are written in a misleading way and can easily be improved.

(unfortunately for the reviewer, lines are still not numbered)

Page 2, “However, it is worth noting that an in silico PCR approach to type MLVA in Brucella from WGS data was recently developed”

Would be more exact to indicate:

“However, it is worth noting that an in silico PCR approach to type MLVA from WGS data was recently developed and evaluated for Brucella”

Along this line of benchmarking, may be worth mentioning https://github.com/Papos92/MISTReSS

Page 3 second line “GeneScan determination was retrieved for Ugandan isolates from published

data [9]”

This sentence is misleading since it suggests that the chromatograms were published as part of the Bwire et al. 2018 publication. However what I understand is that the authors have in the course of the present investigation realized that the MLVA alleles calling published in the 2018 report was incorrect, and have reanalyzed the data. This is a different thing.

Page 3 second paragraph “method proposed by Kendall et al. [10], the formula had to be modified to better fit the sequence length of the motif and the position of the primers (Table 1). It is of note that, the original calculation formula was used for the VC0283 motif but with a modified reverse primer”

Not clear to me. Do the authors just mean that the use of the modified VC0283 reverse primer has no impact on the PCR product size? May be worth recalling the sequence of the previous primer, as in “(AGCCTCCTCAGAAGTTGAG instead of the previous XXXXX)”

“Locus” would be less ambiguous than “motif” (which refers to the repeat unit).

Page 3, page 4 and elsewhere: two designations are used for loci VC0171 (alias VCA0171) and VC0283 (alias VCA0283), please harmonize

Page 4: “returns the number of tandem repeats” and “increasing number (j=2, 3 .., k) of tandem repeats”:

may be clearer to replace “tandem repeats” by “tandemly repeated units”

Page 5 cost analysis paragraph, “Reagents costs for MLVA typing (5 motifs) of V. cholerae isolates were compared using three different methods (Table 5).”

The authors need to expand a little bit. Discuss the relative costs. Recall that Sanger sequencing was used only to produce a reference dataset, but not as a suggestion for routine MLVA typing (as shown, would make no sense in terms of cost).

Table 5

Instead of “Motifs”, the authors probably mean “Loci”?

Cost estimates:

Sanger sequencing: are these commercial costs? Eight dollars for one PCR in terms of reagents and consumables seems a lot.

Genscan-based typing: with only five loci, all five loci should be run in one multiplex PCR. Most laboratories running MLVA on Genscan-type of equipment and significant numbers of strains will multiplex the PCRs. Then the fair reagents cost estimate should be down to one PCR and one run per sample i.e. 11.4 USD.

WGS cost: indicating current best commercial prices (for sequencing 1 strain alone versus as part of a batch of 96) might be useful

Page 5, “However, it should be noted that a modification of either the primer” When running MLVA on a Genscan type of machine, the “formula” is useless. The allele calling software will call each allele base on its associated observed size range. This remark by the authors suggests that the authors are exporting raw size estimates from the Genscan, and then convert by allele calling using the “formula”. This is not the most recommended way to proceed, see available literature.

Page 5, “was applied in order to decrease the number of mismatches, especially those observed

with VC0437 and VCA0283.”

Quote Table 1 (and check locus names, see a previous remark)

Page 7: “hence solving the well-recognized issue of backward compatibility with traditional MLVA typing methods.”

The software is not solving anything! There has never been an issue of backward compatibility when the sequencing reads are longer than the tandem repeat arrays. Recall that when tandem repeats contain internal variations, software for sequence assembly may be able to reconstruct correctly tandem repeats longer than the sequencing reads. Mention alternative approaches (the in silico PCR methods, including in terms of benchmarking https://github.com/Papos92/MISTReSS) and explain why the present approach is believed to be more appropriate at least for V. cholera MLVA.

I would suggest to merge Table 1 and S1 table as S1 table clearly illustrates the impact of the k-mer size.

Discussion: the authors need to comment on the pros and cons of the approach they use here versus the more commonly used in silico PCR approach. In particular they need to indicate that the approach used here work in the Vibrio cholerae context because the VNTRs are unique (so the repeat unit sequence is locus-specific), the tandem repeat arrays are perfect, and there are no indels in the flanking sequences. This is an uncommon situation. They might indicate why they think this approach may be of interest when applicable. Do they think it is because it does not require to have assembled the flanking sequences?

7. PLOS authors have the option to publish the peer review history of their article (what does this mean?). If published, this will include your full peer review and any attached files.

Reviewer #1: No

---

## [Author Response · Author response to Decision Letter 1]

12 Nov 2019

The point-to-point answer to the reviewer has been attached as a word file.

---

## [Editor Report · Decision Letter 2]

14 Nov 2019

Backward compatibility of whole genome sequencing data with MLVA typing using a new MLVAtype shiny application for Vibrio cholerae

PONE-D-19-20834R2

Dear Dr. Ambroise,

We are pleased to inform you that your manuscript has been judged scientifically suitable for publication and will be formally accepted for publication once it complies with all outstanding technical requirements.

With kind regards,

Axel Cloeckaert

Academic Editor

PLOS ONE
---

## [Editor Report · Acceptance letter]

2 Dec 2019

PONE-D-19-20834R2 

Backward compatibility of whole genome sequencing data with MLVA typing using a new *MLVAtype* shiny application for *Vibrio cholerae*

Dear Dr. Ambroise:

I am pleased to inform you that your manuscript has been deemed suitable for publication in PLOS ONE. Congratulations! Your manuscript is now with our production department. 

With kind regards,

on behalf of

Dr. Axel Cloeckaert 

Academic Editor

PLOS ONE